# PSO/GA Combined with Charge Simulation Method for the Electric Field Under Transmission Lines in 3D Calculation Model

**Ru Wang** [†,‡] [ID]**, Jin Tian** *[,‡]**, Fei Wu, Zhenhua Zhang and Haishan Liu**

School of Electric and Electronic Engineering, Shanghai University of Engineering Science, Shanghai 201620, China; paopaojun1995@163.com (R.W.); fei_wu1@163.com (F.W.); zhenhuazhang1985@163.com (Z.Z.); liuhsh@sues.edu.cn (H.L.)

* Correspondence: jintian0120@foxmail.com; Tel.: +86-13764889308
† Current address: Modern Transportation Engineering Center 7921, Longteng Road 333, Songjiang, Shanghai 201620, China
‡ These authors contributed equally to this work.

**Abstract:** The accurate calculation of electric field intensity under transmission lines is more and more important with the expansion of high voltage engineering, which largely determines the site selection and design of transmission projects. In order to get more accurate results, a methodology of a charge simulation method (CSM) combined with an intelligent optimization algorithm for a 3D calculation model is proposed in this paper. Three key points are emphasized for special mention in this optimized charge simulation method (OCSM). First, the number of sub-segments on the finite length conductor, the position and number of the simulating charge set on a sub-segment are taken as the optimization parameters for unified calculation. Secondly, the fitness function of optimization algorithm is constructed by two values, voltage relative error and electric field intensity relative error. Thirdly, a finite element method (FEM) was used to obtain the electric field intensities, which are compared with the results of the proposed algorithm. A simulation case is carried out on a 3D calculation model of 220 kV transmission lines, which verify the effectiveness of the optimization algorithm. The proposed OCSM solves the parameter optimization problem of CSM in the 3D computational model, which considers physical shape of wire span, and has the advantages of strong global search ability and higher calculation accuracy.

**Keywords:** charge simulation method; intelligent algorithm; three-dimension calculation model; transmission lines; finite element method; fitness function

---

## 1. Introduction

With the expansion of the power grid and the rapid improvement of the transmission voltage level, from high voltage (110∼220 kV) to ultra-high voltage (500 kV), the strong electromagnetic field generated by the high voltage transmission lines and transformation system has brought harm to people and the environment. Meanwhile, the problems of electromagnetic field limit value caused by transmission lines have become a key factor affecting the appropriate site selection and design of transmission projects.

The problem of the electric field calculation algorithm is to get closer to real transmission line geometry to ensure the accuracy of calculation. Meanwhile, the problem of calculating an ill-conditioned coefficient matrix generated by a 3D model needs to be solved. Most of the strategies developed aim at: (1) maximizing the accuracy of electric field calculation and (2) balancing optimization accuracy, computational complexity, and calculation time. In this work, particle swarm

optimization (PSO) and genetic algorithm (GA) as optimization algorithms are combined with a charge simulating method in three dimensions. The main contributions of this work are the following:

1. The proposed optimized charge simulation method (OCSM) has higher efficiency and shorter computation time than the finite element method;

2. PSO and GA are used to find the optimal solution that can balance ill-conditioned coefficient matrix and computational accuracy;

3. The OCSM is applied to the calculation of 3D electric field under transmission lines, which consider geometric structure of the line in practical engineering;

4. The accuracy and robustness of proposed algorithm in this paper is greater than normal charge simulation method (CSM).

The remainder of this work is organized as follows: Section 2 presents the most important of the related papers, introducing the electric field calculation algorithm including CSM and finite element method (FEM), focusing on algorithms of optimization CSM. Section 3 presents that 2D and 3D calculation methods based on CSM and the segmented calculation on the finite length wire. Section 4 introduces that the fitness function consisted of two error values and calculation method of fitness value. Section 5 introduces two different optimal algorithm, PSO and GA, including principle and process. In Section 6, calculation model is establish in detail, experimental results are presented, the comparison results illustrate the proposed OCSM is effective and accurate. Section 7 concludes this paper.

## 2. Related Work

Numerical methods for solving electric field are mainly divided into two categories. One is the solution of field boundary problem based on Maxwell equation, the other is to solve it by establishing differential function [1]. The charge simulation method (CSM) is a numerical calculation method of electromagnetic field belonged to the equivalent source. Based on Laplace equation, these hypothetical simulating charges are used to replace the continuous charge distributed on the electrode surface, which can be used to calculate the electric field of the entire field equivalently [2,3]. The finite element method (FEM) is a common numerical method for solving numerical boundary value problems. With the progress of large-scale numerical calculation in recent years, FEM has been gradually applied in the field of engineering electromagnetic field [4–10]. The method of moments (MoM) is suitable for solving complex problems of field and excitation source distribution. It is mainly used in antenna, electromagnetic scattering, and microwave technology.

CSM has been widely used to solve static or quasi-static fields in the past 20 years, which is one of the main algorithms for electrostatic field numerical calculation [11–16]. The various electromagnetic calculation methods combined with CSM are used to accurately calculate the electric field near the transmission lines [17–19]. The various optimization algorithms are used to optimize parameters in CSM such as position of the simulating charges, number of simulating charges [20–30]. The 3D calculation model of overhead transmissions line based on CSM has been established, which regards overhead transmission lines as finite length line charge with sag [31–33]. However, there is no research on optimizing the computational parameters of CSM under the 3D computational model. The existing study of optimization algorithm combined CSM only consider the simulating charge is an infinite length line charge, which cannot obtain accurate calculation results in the real situation.

## 3. Charge Simulation Method Solving Electric Field Near Transmission Lines

According to Maxwell equation and Green's theorem, the equivalent surface charge on the closed surface which surround region $V_1$ can be used to replace the objective field source in the region $V_1$.

The purposed simulation charge method set discrete charges outside the calculated field to replace unknown continuous charges without knowing the charge distribution. The equation set of discrete charges are givens as:

$$[P] \cdot [Q] = [\varphi] \tag{1}$$

where $[P]$ is potential coefficient matrix; $[Q]$ is simulating charges density matrix; $[\varphi]$ is potential matrix.

### 3.1. 2-Dimensional Calculation Method

According to Maxwell equation and Green's theorem, the equivalent surface charge on the closed surface which surround region $V_1$ can be used to replace the objective field source in the region $V_1$.

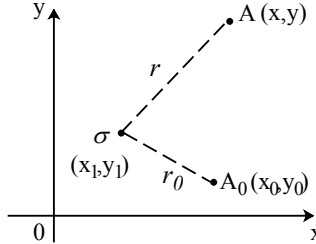

**Figure 1.** The 2D calculation method of infinite line charge.

As shown in Figure 1, when point $A_0$ is the potential reference point, the potential generated by the line charge $\sigma$ at point $A$ in the field is:

$$\varphi = \frac{Q}{2\pi\varepsilon} \ln\left(\frac{r_0}{r}\right) \tag{2}$$

where

$$r = \sqrt{(x - x_1)^2 + (y - y_1)^2}$$

$$r_0 = \sqrt{(x_1 - x_0)^2 + (y_1 - y_0)^2}$$

### 3.2. 3-Dimensional Calculation Method

The 2D model is simple and rough, which cannot calculate power frequency electric field accurately. Hence, a 3D calculation method based on CSM is proposed.

The overhead transmission lines are regarded as an ideal flexible cable chain because of the span of several hundred meters. Therefore, the transmission lines which are hung between two towers present the shape of catenary and the simplified 3D calculation model of overhead transmission lines is constructed and shown in Figure 2.

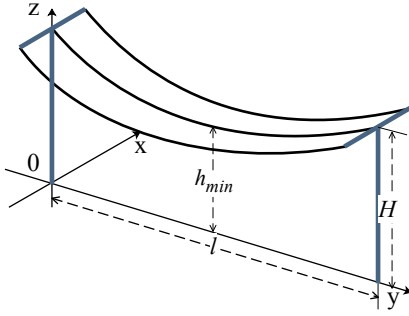

**Figure 2.** The 3D calculation model of overhead transmission lines.

In order to obtain the geometric shape of overhead transmission lines, the material and physical information of transmission line are used for calculation. The equation of catenary is:

$$h_{\min} = \frac{\gamma}{g}\left[1 - \cosh\left(\frac{l}{2}\frac{g}{\gamma}\right)\sqrt{1 + \left(\frac{h}{L_h}\right)^2}\right] + \frac{h}{2} \tag{3}$$

where

$$L_h = \frac{2\gamma}{g} sh\left(\frac{gl}{2\gamma}\right)$$

In (3), $h_{min}$ is the coordinate of the lowest point in the Z direction; $l$ and $h$ are horizontal distance and vertical distance between two hanging points respectively; $\gamma$ is horizontal stress of conductor; $g$ is the specific load of the conductor.

When both of the hanging points are of equal height, the sag of conductors are calculated by the catenary equation:

$$z = h_{\min} \cdot \cosh\frac{y}{h_{\min}} = a\cosh\frac{y}{a} \tag{4}$$

where $z$ is the coordinate of a point on the catenary in the Z direction, $a$ is catenary coefficient, and $y$ is the coordinate of a point on the catenary in the Y direction.

It is assumed that all overhead transmission lines of the same conductor type hanging the same tower have the same shape of catenary under the same operating environment, the 3D calculation model are optimized as follows:

In the Figure 3, $\dot{Q}_n$ and $-\dot{Q}_n$ are the $n$th transmission lines simulating charge and mirror simulating charge respectively. The potential generated by line charge at point A is:

$$\dot{\varphi}_n = \frac{1}{4\pi\varepsilon_0}\int_s \dot{Q}_n\left(\frac{1}{\rho_n} - \frac{1}{\rho_n'}\right)ds_n \tag{5}$$

where $\varepsilon_0$ is permittivity of vacuum, $l$ is conductor length,

$$\rho_n' = \sqrt{(x-x_n)^2 + (y-y_n)^2 + (z+z_n)^2}$$

$$\rho_n = \sqrt{(x-x_n)^2 + (y-y_n)^2 + (z-z_n)^2}$$

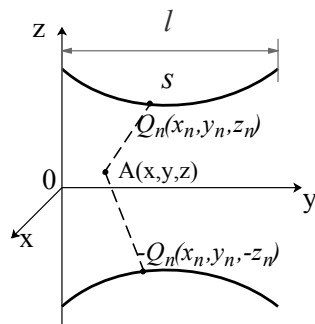

**Figure 3.** The 3D calculation method of finite nonlinear line charge.

Based on (4), the conductor length element can be written as:

$$ds_n = \sqrt{1 + z_n'^2}dy_n = \cosh\frac{y_n}{a}dy_n \tag{6}$$

The potential coefficient generated by line charge at point A is:

$$P_n = \int_0^l \left(\frac{1}{\rho_n} - \frac{1}{\rho_n'}\right)\cosh\frac{y_n}{a}dy_n \tag{7}$$

where:

$$\rho_n = \sqrt{(x - x_n)^2 + (y - y_n)^2 + \left(z - a\cosh\frac{y_n}{a}\right)^2}$$

$$\rho_n' = \sqrt{(x - x_n)^2 + (y - y_n)^2 + \left(z + a\cosh\frac{y_n}{a}\right)^2}$$

Because the potential coefficient $P_n$ of the $n$th transmission line is an integral function of the line length, the discrete integral equation is used to solve $P_n$. The $n$th transmission line of length $l$ are evenly divided into $M$ segments and simulating line charge $\dot{Q}_n^m$ is set in each segment. When the length of each segment transmission line is small enough, the integral function can be obtained by numerical calculation. According to the electric field superposition principle, the potential generated by $M$ sub-segment simulating charges and $N$ transmission lines at any point $A$ in space is:

$$\dot{\varphi} = \sum_{n=1}^{N}\sum_{m=1}^{M}\frac{1}{4\pi\varepsilon_0}p_n^m\dot{Q}_n^m \tag{8}$$

where

$$p_n^m = \left(\frac{1}{\rho_n^m} - \frac{1}{\rho_n^{m'}}\right)\cosh\frac{\left(y_n^m - y_n^{m-1}\right)}{2a}\frac{l}{M}$$

## 4. Fitness Function

In the charge simulation method, the calculation of electric field intensity depends only on the position and number of simulating charges under the same example model. It also depends on the number of conductor segments M in the 3D calculation of power-frequency electric field. In order to find the optimal position, number of simulating charge, and number of M sub-segments, the fitness function is establish.

The fitness function consists of two error values, which are added by different weights. The two calculation error values are separately based on the potential difference between check points and real match points and electric field intensity difference between the calculation results of CSM and FEM.

### 4.1. The Relative Calculation Error between Check Points and Match Points

The arrangement of simulating charges, match points, and check points in transmission conductors on the cross section is shown in Figure 4.

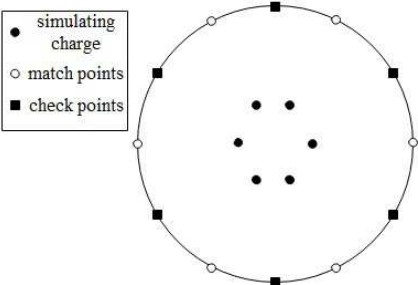

**Figure 4.** Distribution of the simulating charges, match points, and check points.

Assuming that $T$ simulating charges are set inside each phase conductor ($T = 6$ in Figure 4), the match points are set on the surface of the conductor and the number of match points equals the number of simulating charges. Some other check points are taken to verify the calculation results which are different from the position of the matching point on the surface of the conductors. The value

of simulating charge $\dot{Q}_{n,t}^m$ was calculated from the potential at the matching points, then the potential at the check point was calculated by:

$$\begin{cases} [\dot{Q}_n^m] = \sum\limits_{t=1}^{T} \sum\limits_{m=1}^{M} \sum\limits_{n=1}^{N} [P_n^m]^{-1} [V_n^m] \\ [V_{check}] = \sum\limits_{t=1}^{T} \sum\limits_{m=1}^{M} \sum\limits_{n=1}^{N} [P_{check}] [\dot{Q}_n^m] \end{cases} \tag{9}$$

The calculation error value of check points used for optimization algorithms is given by the following equation:

$$CE_1 = \frac{1}{S} \sum\limits_{s=1}^{S} \left| \frac{V_{check,s} - V_n}{V_n} \right| \tag{10}$$

where $S$ is the number of all simulating charges of each segment of each transmission conductor, $V_n$ is potential of the $n$th transmission conductor, and $V_{check,s}$ is the potential of the $s$th check point.

*4.2. The Calculation Error between CSM and FEM*

The three-dimensional calculation of CSM consider of transmission line sag so that calculation accuracy is greatly related to the number of sub-segments. With the larger number of sub-segments $M$, the numerical calculation method becomes more accurate, which is to solve the integral function. However, too many sub-segments brings a huge amount of simulating charges, which lead to generate ill-conditioned coefficient matrix, decrease robustness and the increase of calculation time. In order to obtain the optimal segment number $M$, FEM, a numerical calculation method commonly used in the field of power-frequency electromagnetic field, is used to calculate the electric field intensity for checking the calculation results of CSM.

By using software ANSYS, 3D calculation model is established, boundary conditions and potential excitations are set, adaptive grid subdivision is carried out, and solution equations are established. The calculation error value between FEM and CSM under the same calculation model which is used for optimization algorithms can be written as:

$$CE_2 = \frac{1}{I} \sum\limits_{i=1}^{I} \left| \frac{E_{FEM,i} - E_{CSM,i}}{E_{CSM,i}} \right| \tag{11}$$

where $I$ is the number of all calculation points in space, $E_{FEM,i}$ is the electric field intensity of FEM on space point $i$, $E_{CSM,i}$ is the electric field intensity of CSM on space point $i$.

**5. Optimized Charge Simulation Method**

The intelligent optimization algorithm can solve multi-objective and nonlinear problems, and can also find the optimal solution in the global scope. In order to obtain the optimal parameters of three-dimensional calculation of CSM, which is the optimal simulating charge number, position, and number of sub-segment $M$. The features of OCSM to find optimal parameters: (1) it contains a large number of sparse coefficient matrices; (2) searching range for optimization parameters is from 1 to 1000; (3) the optimal solution is determined by the fitness function value of each parameter. Therefore, the selected optimization algorithm needs to have excellent global search ability within a wide range of parameters (1~1000). The selected optimization algorithm needs to contain implicit parallelism, which can improve the performance and efficiency of the sparse matrix operations. The selected optimization algorithm should be convenient to embed fitness function calculation and easy to implement under programming. In order to meet the above all requirements, PSO and GA are selected from all optimization algorithms to solve the problem respectively.

The program flow chart of the OCSM is shown in Figure 5. The CSM is encapsulated as a function of simulation charge parameters. Meanwhile, the simulation charge parameters are swarm parameter in PSO/GA. The function (CSM) is called in the program, and returned result is the relative voltage error value and the relative electric field intensity error value obtained by CSM. The fitness function is calculated based on the returned results. Through the optimization principle of PSO/GA, the swarm individuals with good fitness are left and the swarm individuals with poor fitness are updated. Until the set number of iterations is satisfied, the program stops and outputs the optimal parameters and the optimal solution.

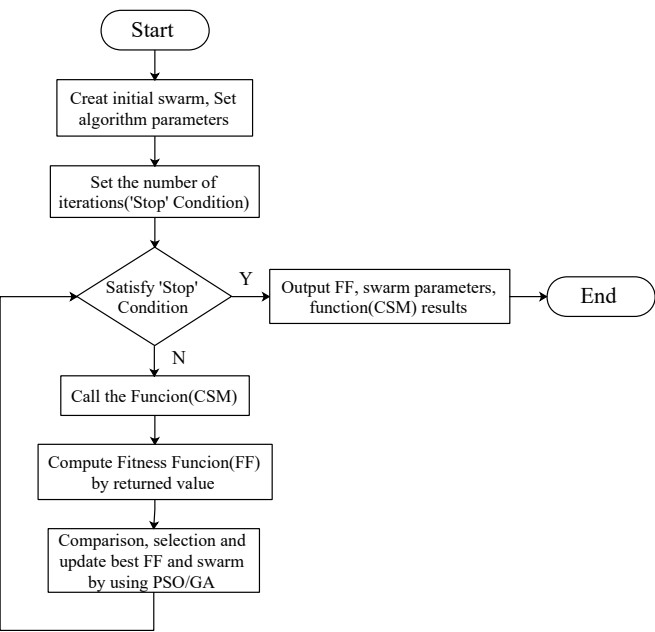

**Figure 5.** The flow chart of the optimized charge simulation method.

## 5.1. Particle Swarm Optimization

The particle swarm optimization (PSO) algorithm is easy to implement without gradient information, which is applied to discrete optimization problem with good effect. There is a swarm of $m$ particles moves at a certain speed in the D-dimensional search space. When each particle searches, it considers the best historical point that it can find and the best historical point of other particles in the swarm, then it changes its position (state, i.e., solution) on this basis.

Assuming that $x_i$ is the position of the $i$th particle, $v_i$ is the velocity of the $i$th particle, $p_i$ is the the best point in history that the $i$th particle has experienced, $\boldsymbol{p}_g = (p_{g1}, p_{g2}, \cdots, p_{gD})$ is the best point in history that all particles in the swarm have experienced. The position and velocity of particles are changed according to the following equation:

$$\begin{cases} v_{iD}^{k+1} = v_{iD}^k + c_1 \xi \left( p_{iD}^k - x_{iD}^k \right) + c_2 \eta \left( p_{gD}^k - x_{iD}^k \right) \\ x_{iD}^{k+1} = x_{iD}^k + v_{iD}^{k+1} \end{cases} \tag{12}$$

where $c_1, c_2$ is learning factor, which make the particle have the ability to learn so that it can close to its own historical best point and the historical best point in swarm; $\xi$ and $\eta$ are uniformly distributed random numbers between 0 and 1. The velocity of particle is limited within the maximum velocity $v_{max}$. In order to improve the convergence efficiency of the algorithm, Shi and Eberhart presented the concept of inertia weight and modified the speed update formula as follows:

$$v_{id}^{k+1} = \omega v_{id}^k + c_1 \xi \left( p_{id}^k - x_{id}^k \right) + c_2 \eta \left( p_{gd}^k - x_{id}^k \right) \tag{13}$$

where $\omega$ is inertia weight, which determines the credibility of the current velocity of particles. The appropriate $\omega$ can obtain the local optimal solution and the global optimal solution [34,35].

The parameters of PSO algorithm, which control algorithm, are set as follows:

1. Swarm size $N$, which affects the search ability and computation amount of the algorithm, are generally set from 20 to 40;

2. The range $R$ of particles is determined by the optimization problem;

3. The maximum velocity $V_{max}$, which determines the maximum distance the particle moves each time, is generally set from 10% to 20%;

4. The inertia weight $\omega$ controls the influence of the former velocity on the current velocity. When w is between (0.8–1.2), the algorithm has a faster convergence speed;

5. The acceleration coefficients $c_1$ and $c_2$ represent the acceleration weights of the particle to its own extreme value and global extreme value respectively . $c_1$ and $c_2$ are usually equal to 2.0;

6. The stop condition controls the end of the algorithm. Generally, the maximum number of cycles is set as the stop condition.

## 5.2. Genetic Algorithm

The genetic algorithm (GA) constructs a fitness function based on the objective function of the problem, and then evaluates, genetic calculates and selects the population that is composed of multiple solutions. The individual with the best fitness value is obtained as the optimal solution of the problem after multiple generations of reproduction.

GA is intelligent Optimization algorithm based on swarm, so it usually adopts random method or specific method to construct an initial swarm. The fitness function is used to represent the living environment adaptability of each individual in the swarm in GA. The fitness value is generally determined according to the optimized objective function, and is the only deterministic index of the individual survival chance in the swarm. The larger the fitness value of an individual, the greater the probability that the individual will be inherited to the next generation. Selected individuals reproduce in pairs to form new swarm. After the process of selection and reproduction is repeated many times, the solution of the individual with the best fitness value is considered as the optimal solution of the problem [36,37]. GA flow chart is shown as Figure 6.

The parameters of genetic algorithm, which control algorithm, are set as follows:

1. Swarm size, which affects the search ability and computation amount of the algorithm, is generally set from 20 to 100;

2. The optimal solution interval is determined by the optimization problem in this paper;

3. The number of termination evolution of genetic algorithm, which balances algorithm convergence and computation time, is generally set from 100 to 500;

4. The variation probability in the algorithm guarantees the diversity of the swarm. The probability of variation generally ranges from 0.001 to 0.2;

5. The crossover probability in the algorithm can effectively update the swarm. The crossover probability is usually set from 0.4 to 0.99.

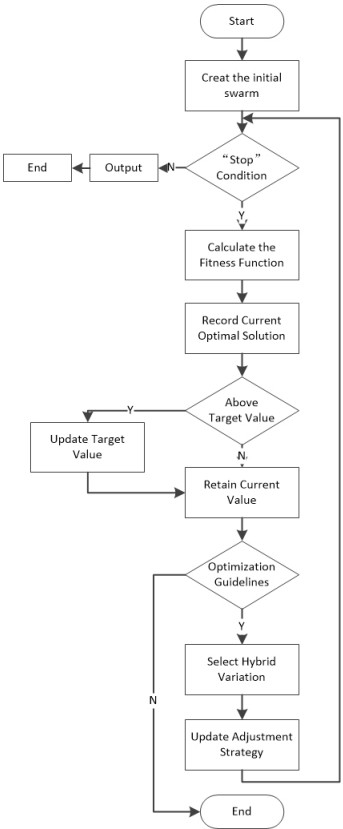

**Figure 6.** The flow chart of the genetic algorithm.

## 6. Results and Discussions

### 6.1. The Analysis of Calculation Example

There is a calculation example of the 220 kV transmission line shown in Figure 7, where height of transmission lines to the ground *H* is 12 m, distance between each line *D* is 5 m, span of transmission lines *l* is 200 m. Assuming that the type of transmission lines is 2 × LGJ-300/40, the diameter of sub-conductors is 23.94 mm, the distance of two sub-conductors is 400 mm.

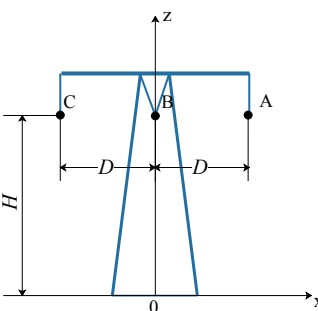

**Figure 7.** Example of the 220 kV transmission line.

The electric field intensity vector of any space point (*x*, *y*, *z*) below the transmission conductors is:

$$\dot{E} = -\left(\frac{\partial \dot{\varphi}}{\partial x} \cdot e_x + \frac{\partial \dot{\varphi}}{\partial y} \cdot e_y + \frac{\partial \dot{\varphi}}{\partial z} \cdot e_z\right) \tag{14}$$

where $e_x$, $e_y$, $e_z$ are unit vector of the electric field in the $x$, $y$, and $z$ directions. Because the electric field component $E_z$ is much larger than $E_x$ and $E_y$, the electric field intensity effective value $E$, which ignore the electric field intensity in the $x$ and $y$ directions, can be given as follows:

$$\begin{cases} \mathbf{E} = \mathbf{F} \cdot \mathbf{Q} \\ f_{t,m} = \frac{l}{4\pi\varepsilon M} \cosh \frac{y_m - y_{m-1}}{2a} \left[ \frac{z_t - z_{t,m}}{(d_{t,m})^3} - \frac{z_t + z_{t,m}}{(d'_{t,m})^3} \right] \end{cases} \tag{15}$$

where $\mathbf{F}$ is electric field intensity coefficient matrix, $f_{t,m}$ is $t$ row, $m$ column matrix element of matrix $\mathbf{F}$, $z_t$ is the height of the $t$th calculation point of space, $z_{t,m}$ is the height of $m$th simulating charge, $d_{t,m}$ is distance between the $t$th calculation point of space and the $m$th simulating charge, $d'_{t,m}$ is distance between the $t$th calculation point of space and the $m$th mirror charge.

## 6.2. Calculation Results of CSM and FEM

The 3D calculation method results of CSM and FEM are shown in Figure 8, Figure 9 separately when the calculation height of electric field intensity is 2 m.

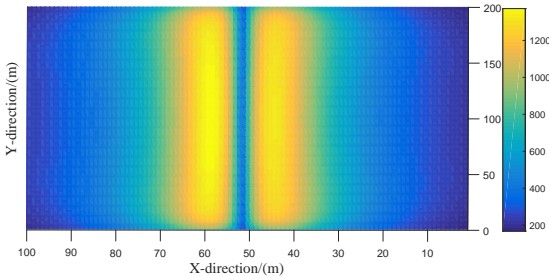

**Figure 8.** The 3D calculation method results of charge simulation method (CSM).

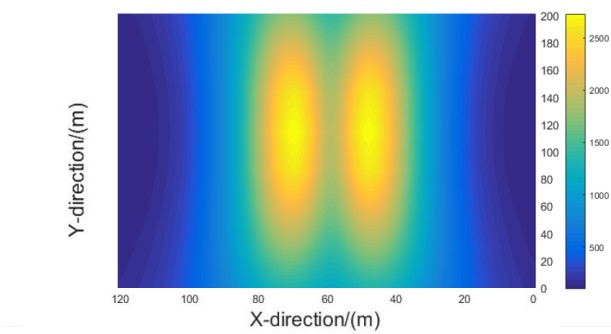

**Figure 9.** The 3D calculation method results of finite element method (FEM).

According to the comparison between Figures 8 and 9, it is obvious that the calculation results of electric field intensity have similar saddle-shaped distribution in three-dimensional model by two methods. The 2D calculation method only shows the variation characteristics of the electric field intensity between three transmission lines. However, the 3D calculation method takes into account the change of the ground height of the conductor, which shows that the electric field intensity increases with the increasing of lines sag.

It is the distribution of electric field intensity on different observation planes, as shown in Figure 10. Where *2D-CSM* is result of observation surface $y = 0$ m, $z = 2$ m; *3D-CSM1* is result of observation surface $y = 0$ m, $z = 2$ m based on 3D calculation method; *3D-CSM2* is result of observation surface $y = 100$ m, $z = 2$ m based on 3D calculation method. The comparison between 2D and 3D calculation

method illustrates the physical shape of the conductor has a great influence on the calculation of the maximum electric field intensity and electric field distribution.

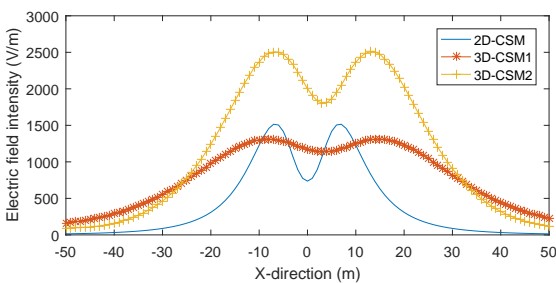

**Figure 10.** Comparison diagram of electric field intensity.

*6.3. Optimization Results and Analysis*

The most important problem is how to determine the position and number of the simulating charges. First, the optimization number and position of simulating charges on a cross section of a bundle of wires are calculated based on 2D calculation model. Then, the three optimal parameters, number and positon of simulation charges, number of sub-segments, are calculated based on 3D calculation model.

The calculation parameters of the two optimization algorithms are shown in Table 1. It obvious shows that the number of iteration of PSO an GA both are 100, which balances the computation time and algorithm convergence. The swarm size of PSO are generally set between 20 and 40 and swarm size of GA are generally set between 20 and 100. The swarm size is chosen to be 30 considering the complexity of calculation in paper. The generation value of the optimal solution for segment number $M$ is $1{\sim}1000$, so the solution limit is set as (1,1000). According to experience and experiments, when the acceleration coefficient $c_1$ and $c_2$ of PSO are 0.2 and the weight factor of PSO is 0.8, the algorithm can achieve excellent results. By running and debugging, it obvious obtain that when GA variation probability and crossover probability are 1% and 0.9% respectively, the algorithm has high convergence and optimization efficiency.

**Table 1.** Particle swarm optimization (PSO) and genetic algorithm (GA) algorithm parameters.

| CSM + Algorithms | Parameters |
|---|---|
| | iteration = 100 |
| PSO | swarm size = 30, limit = (0,1000) |
| | acceleration coefficients $c_1 = c_2 = 2$, weight = 0.8 |
| | iteration = 100 |
| GA | population size = 30, limit = (0,1000) |
| | variation probability = 1%, crossover probability = 0.9 |

For multiple cycles in algorithm, suppose the time complexity of the loop body is O(n), and the number of cycles of each cycle is a, b, c..., then the time complexity of cycles is O (n × a × b × c...). Therefore, according to Table 2, the time complexity of PSO and GA both are T(n) = O (30 n). And the time complexity of PSO and GA both can be written as T(n) = O(n).

**Table 2.** The relationship between the relative voltage error and number and position of the simulating charges.

| Simulating Radius | The Number of Simulating Charges | The Relative Error of Voltage |
|---|---|---|
| $0.15r < r_{sim} < 0.9r$ | 1~5 | $0.03 > \delta > 0.001$ |
| | 6~20 | $10^{-4} > \delta > 10^{-12}$ |
| | 20~30 | $10^{-6} > \delta > 10^{-8}$ |
| | 30~36 | $0.4 > \delta > 0.001$ |
| $0 < r_{sim} < 0.15r$ and $0.9r < r_{sim} < r$ | 1~5 | $0.01 > \delta > 0.0012$ |
| | 6~20 | $10^{-4} > \delta > 10^{-5}$ |
| | 20~30 | $10^{-4} > \delta > 10^{-5}$ |
| | 30~36 | $0.8 > \delta > 10^{-3}$ |

The fitness function in intelligent optimization algorithms is computation accuracy is given below:

$$FF = CE_1 \cdot 0.5 + CE_2 \cdot 0.5 \tag{16}$$

The value of $CE_1$ determines the accuracy of coefficient matrix operation, and the value of $CE_2$ determines the accuracy of electric field intensity results.

### 6.3.1. Optimized Parameters Based on 2D Calculation Model

According to Figure 4, assuming that $T$ simulating charges are set on a cross section of a bundle of wires, the match points and check points are set on the surface of conductor and the number of match points and check points equal to the number of simulating charges. Supposing the number of simulating charges is 1~36 and range of simulating charges radius is 0~r, the relative errors of voltage under different conditions are calculated respectively.

As shown in Table 2, when simulating radius is between $0.15r$ and $0.9r$, the relative error of voltage is less than when simulating radius is between $0.15r$ and $0.9r$. No matter how much radius, the relative error of voltage is the smallest, which is between $10^{-4}$ and $10^{-12}$, when the number of simulated charges is 6~20. Based on data in Table 2, it illustrate that when the number of simulated charge is too large or too small, and the position of simulated charge is too close to the wire surface, the calculation matrix is close to the singular value, which makes the relative error of voltage improve and the calculation accuracy decreases. When the number of simulated charges is 6~20 and the simulated radius is $15r$~$0.9r$, we can obtain the best calculation accuracy.

It can be proved more clearly from Figures 11 and 12 that the relative error of voltage get a smaller value when the number range of simulating charge is 5~20 and the position range of simulating charges is $0.2r$~$0.5r$. The optimal parameters calculated are that the number of simulated charges is 13 and the position coefficient of simulated charges is 0.25.

In calculation example shown in Figure 7, on a cross section of a bunch of wires, the equivalent radius of the conductor is 0.2 m, relative error of voltage is the smallest, which is $1.2871 \times 10^{-12}$, when the number of simulating charges is 13 and the simulating charges are distributed on a circle with a radius of 0.05 m.

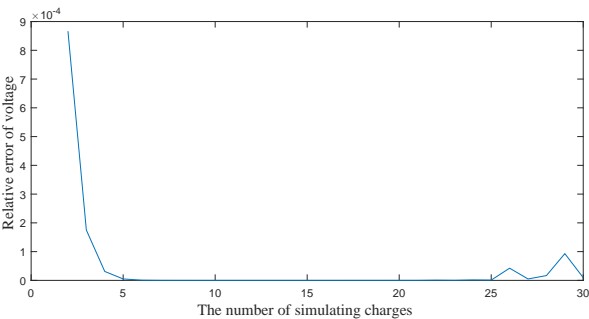

**Figure 11.** The relationship between number of simulating charges and relative error of voltage.

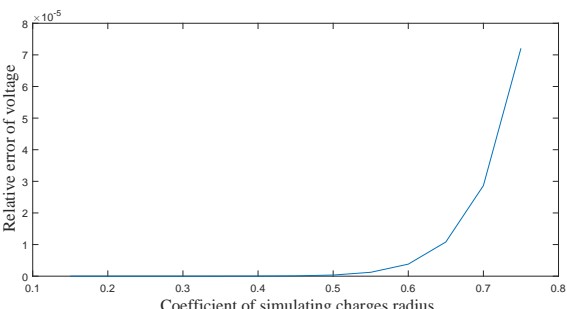

**Figure 12.** The relationship between position of simulating charges and relative error of voltage.

6.3.2. Optimized Parameters Based on 3D Calculation Model

In the 3D calculation method based on CSM, the conductor with length *l* is divided into *M* sub-segments, and the simulating charges are set on each sub-segment, which is to solve the integral problem by numerical calculation method. The larger the sub-segments number *M* is, the more accurate the result of integral solution will be. Meanwhile, it will also lead to the increase of calculation time and generate ill-conditioned coefficient matrix, which make relative error of voltage increase and the electric field intensity be distorted. Based on calculation example shown in Figure 7, two optimization algorithms are combined with CSM for 3D electric field calculation respectively.

The search processes of PSO and GA with optimized number *M* of sub-segments after 100 iterations are represented in Figure 13. The optimal number of sub-segments *M* is 332 through two intelligent optimization algorithms, which minimizes the calculation error consisted of the relative error of voltage and the relative error of electric field intensity.

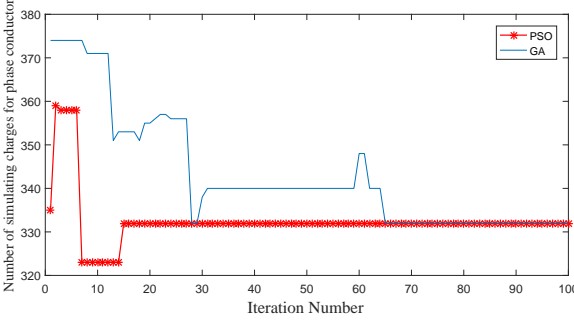

**Figure 13.** The relationship between iteration number and number *M* of sub-segments.

As shown in Table 3, PSO and GA combined with CSM were used respectively to calculate the optimal parameters with different numbers and positions of simulating charges on the cross section of

conductors. By comparing calculation results as shown in Table 3, it is obvious that both PSO and GA get the optimal result that $M$ is 332 and the number of simulating charge is 1 on the cross section. When the number of simulating charges on the cross section increases to 2, the singular value in the coefficient matrix is generated, which sharply increases the relative errors of voltage (from $8.1783 \times 10^{-4}$ to 0.4392) and electric field intensity (from 14.98% to 41.09%). If the number of simulating charges on the cross section is greater than 2, the exact calculation results cannot be obtained.

**Table 3.** Optimized results of PSO and GA.

| CSM +Algorithms | Number and Position of Simulating Charges on a Cross Section of Conductor | The Number M of Sub-Segments | Relative Error of Voltage | Relative Error of Electric Field |
|---|---|---|---|---|
| | num = 1, radius = 0.25$r$ | 332 | $8.1783 \times 10^{-4}$ | 14.98% |
| | num = 2, radius = 0.25$r$ | 415 | 0.4329 | 41.09% |
| PSO | num = 3, radius = 0.25$r$ | NaN | NaN | NaN |
| | ⋮ | ⋮ | ⋮ | ⋮ |
| | num = 1, radius = 0.25$r$ | 332 | $8.2003 \times 10^{-4}$ | 14.83% |
| | num = 2, radius = 0.25$r$ | 416 | 0.4060 | 43.21% |
| GA | num = 3, radius = 0.25$r$ | NaN | NaN | NaN |
| | ⋮ | ⋮ | ⋮ | ⋮ |

*6.4. Experiment Verification and Conclusion*

In order to verify the effectiveness of the above OCSM in calculating the electric field of the transmission line, general CSM and FEM are used for verification. The experiment calculation model, as shown in Figure 7 above, is the 220 kV three-item transmission lines. The three electric field calculation algorithms shown in Table 4 are all running on 1.6 GHz, 8 GB RAM, and Intel i5-8250 CPU computer. The OCSM and CSM both run on MATLAB 2016b, FEM run on Ansoft Maxwell 15.0. The software Maxwell divides the solving field adaptively and achieves very high accuracy in finite element method. Therefore, the results of FEM are used as comparison standard to calculate the relative error of electric field intensity of OCSM and general CSM.

**Table 4.** Comparison of experiment results of optimized charge simulation method (OCSM), charge simulation method (CSM), and finite element method (FEM).

| Calculation Algorithm | OCSM | CSM | FEM |
|---|---|---|---|
| Computation Time(seconds) | 4257.505 | 2.073 | 84096.911 |
| Relative error of voltage | $8.1614 \times 10^{-4}$ | $5.2213 \times 10^{-4}$ | / |
| Relative error of electric field intensity | 14.905% | 55.317% | / |

It is obvious from Table 4 that the OCSM takes much longer to compute than CSM because of its principle of iteratively searching for the optimal solution. The FEM obtain very accurate calculation values, but the modeling is complex and the calculation time is up to 84096.911 s. The relative voltage error is the result of accuracy test through setting the check point of simulation charge in the OCSM and CSM. The experiment results show that the relative voltage error of OCSM is slightly higher than that of CSM. Because the relative voltage errors of two methods both are within the range of $10^{-4}$,

the matrix and numerical calculation in the algorithm is accurate. Meanwhile, the relative electric field error is reduced from 55.317%to 14.905% which indicates that the computation accuracy and numerical efficiency of OCSM is greatly improved compared with the general CSM.

The calculation results of the three methods for the electric field intensity under the transmission line at the observation plane $Z = 2$ m are represented in Figure 14. The general CSM cannot represent the effect of conductor sag on electric field intensity. In order to compare the calculation results of the three methods more clearly, the calculation results of observation plane $Y = 100$ m and $Z = 2$ m were represented, as shown in Figure 15. It is obvious shown that the electric field intensity distribution trend of OCSM is almost consistent with that of FEM and calculation results of general CSM are quite different from those of FEM. It also be seen from Table 4 that the relative result error between CSM/OCSM with FEM are 55.317% and 14.905% respectively.

For the study of electric field intensity under transmission line, the most important calculation region is x = [−10, 10] under transmission lines. The maximum value and distribution characteristics of electric field intensity fluctuate strongly in this region. In the region x = [−10, 10], the relative error of the calculated results between OCSM and FEM is 1.494%, which indicates OCSM has a high computation accuracy. In the region where x < −10 and x > 10, since the electric field intensity is rapidly decreasing and the distribution trend of electric field is basically the same, it is of little significance to calculate the value and distribution of environmental electric field.

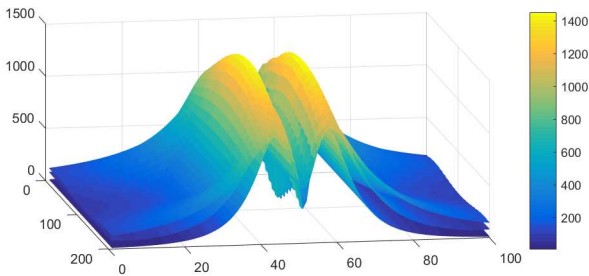

**Figure 14.** 3D view of electric field intensity of CSM, OCSM, and FEM.

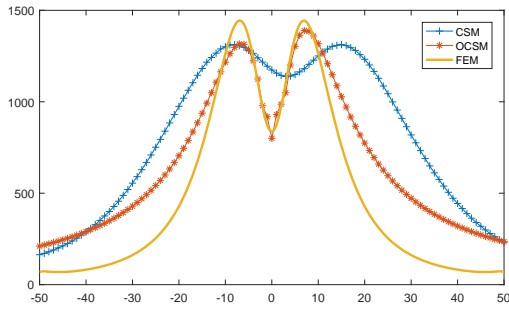

**Figure 15.** Electric field intensity of CSM, OCSM, and FEM at observation plane $Y = 100$ m, $Z = 2$ m

## 7. Conclusions

In this paper, the 3D calculation methods based on CSM are analyzed to calculate the electric field intensity under a conductor, which is seen as a naturally drooping wire with finite length charges. In order to obtain the optimization parameters related to accuracy and robustness in a 3D calculation method, which is the simulated number of charge, position, and sub-segments number, the calculation method of intelligent optimization algorithm combined with CSM was proposed. PSO and GA are respectively used to verify the non-contingency of the optimization results. Meanwhile, based on FEM, a 3D conductor model was built by Maxwell software, and the electric field intensity under the line

was calculated. Not only is the relative error of voltage on checkpoints taken as the fitness function of the optimal algorithm, the electric field intensity result comparison between CSM and FEM is also taken as the fitness function.

The CSM combined with PSO/GA algorithm was used to find the optimal parameters based on a 220 kV transmission line calculation model. From the results, it is clear that PSO and GA are adopted for global optimization. The fitness function includes two relative errors of voltage and electric field intensity in the optimization algorithm and ensures high results accuracy and a good calculation of robustness in the proposed algorithm. Electric field intensity calculated by the intelligent optimization algorithm is also compared with the solving result based on FEM. The comparison shows the same electric field distribution characteristics and the calculated results are very close.

By combining two different optimization algorithms, establishing a three-dimensional computing model, and using FEM, the calculation of electric field intensity based on OCSM is more accurate and more suitable for practical engineering environments. The relative error of voltage obtained is $8.2 \times 10^{-4}$, and the obtained relative error of electric field intensity is 14.905%. The calculation times of OCSM and FEM are 1.183 h and 23.36 h. The result strongly proves the time of OCSM proposed in this paper is shorter than FEM and the accuracy of the algorithm is higher than normal CSM. The OCSM proposed in this paper is more suitable for transmission lines under the 3D computing model, and has the characteristics of strong global search ability, effectiveness, and high calculation accuracy.

**Author Contributions:** The authors' individual contributions are provided as follows: Conceptualization, R.W., J.T., F.W., Z.Z. and H.L.; methodology, R.W., J.T. and F.W.; software, R.W.; validation, R.W., J.T. and F.W.; resources, R.W., Z.Z. and H.L.; writing–original draft preparation, R.W.; writing–review and editing, R.W., J.T., F.W., Z.Z. and H.L.; supervision, Z.Z. and H.L.; project administration, J.T.; funding acquisition, J.T. and F.W.

**Funding:** This research was funded by National Natural Science Foundation of China grant number 61272097, Shanghai Science and Technology Development Foundation grant number 15ZR1418900 and Shanghai Science and Technology Development Foundation grant number 18511101600.

**Conflicts of Interest:** The authors declare no conflict of interest. The funder had no role in the design of the study; in the collection, analyses, or interpretation of data; in the writing of the manuscript, or in the decision to publish the results.

## Abbreviations

The following abbreviations are used in this manuscript:

| | |
|---|---|
| PSO | Particle Swarm Optimization |
| GA | Genetic Algorithm |
| 3D | Three Dimension |
| CSM | Charge Simulation Method |
| OCSM | Optimized Charge Simulation Method |
| FEM | Finite Element Method |

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
