# Peer review of "PSO/GA Combined with Charge Simulation Method for the Electric Field Under Transmission Lines in 3D Calculation Model"

_electronics, doi:10.3390/electronics8101140_

Round 1

Reviewer 1 Report

This paper presents an optimization algorithm combined with charge simulation method for the electric field under transmission lines. The topic is interesting and the approach is worth consideration, but I have some concerns:

Comment 1:
In Page 7, Fig. 5, the flowchart of the genetic algorithms must be clearly established in the paper.

Comment 2:
The proposed an approach was applied in the test system. How it will be applied in the field test in the future.

Comment 3:
The conclusion section is weak. It just presents that the authors have accomplish a task verified by test results, while the novelty or specialty of this work is not stressed.

Comment 4:
In References, According to the manuscript style, some contents of this paper need to be revised.

Author Response

Reviewer can check a point-by-point response to comments in "covering letter1". Reviewer can see the revised manuscript in "Revised paper".

Reviewer 2 Report

In this paper, authors present the optimization algorithm for the electric field under transmission lines. Authors should address the following issues:

The title is too general, authors should highlight the optimization method name in the title.

Section with the number zero is unusual. Numbering should be from 1.

In the introduction section, I would expect research hypothesis, which authors will justify within the paper.

The last paragraph, authors should briefly describe individual chapters.

The second chapter should be devoted to the state of the art, where authors describe the recent research in this area instead of mixing it in the introduction section.

Authors should justify why they use PSO and GA algorithms in the comparison with others evolutionary algorithms.

The figure 5 should be rewritten to be more particular. Individual blocks should clearly reflect the parts of GA.

In the both algorithms, authors should clearly describe the parameters, controlling the optimization algorithms.

Authors should justify using parameters in the table 2.

It would be surely worth adding the time complexity of the optimization methods.

Authors should compare the proposed method with other evolutionary methods, as well as they should provide analysis of the optimization parameters regarding the results of the simulation.

Many of the references are not up to date. These should be replaced with recent literature.

Author Response

Reviewer can check a point-by-point response to the comments in "covering letter2". Reviewer can see the revised manuscript in "Revised paper".

Round 2

Reviewer 1 Report

The comments has be answered in the paper.

Reviewer 2 Report

Authors have addressed all the my comments.